# Genome dependent Cas9/gRNA search time underlies sequence dependent gRNA activity

E. A. Moreb 🅿 [1] & M. D. Lynch 🅿 [1✉]

CRISPR-Cas9 is a powerful DNA editing tool. A gRNA directs Cas9 to cleave any DNA sequence with a PAM. However, some gRNA sequences mediate cleavage at higher efficiencies than others. To understand this, numerous studies have screened large gRNA libraries and developed algorithms to predict gRNA sequence dependent activity. These algorithms do not predict other datasets as well as their training dataset and do not predict well between species. Here, to better understand these discrepancies, we retrospectively examine sequence features that impact gRNA activity in 44 published data sets. We find strong evidence that gRNA sequence dependent activity is largely influenced by the ability of the Cas9/gRNA complex to find the target site rather than activity at the target site and that this drives sequence dependent differences in gRNA activity between different species. This understanding will help guide future work to understand Cas9 activity as well as efforts to identify optimal gRNAs and improve Cas9 variants.

[1] Department of Biomedical Engineering, Duke University, Durham, USA. ✉email: michael.lynch@duke.edu

Since their discovery in 2012, Clustered Regularly Interspaced Short Palindromic Repeats (CRISPR) systems have revolutionized how we manipulate biology[1]. CRISPR-Associated Protein 9 (Cas9), from *S. pyogenes*, was the first CRISPR system characterized and enables targeted cleavage of double-stranded DNA[1]. The successful application of CRISPR systems is dependent on a given guide RNA (gRNA) but understanding which gRNA sequences effectively cleave their targets has proven challenging[2–4]. Predictive algorithms have been developed to select gRNA with improved on-target activities[3,5–10]. These algorithms rely on sequence features of the gRNA and target site. While many of these algorithms have achieved good predictability within their training data, predictions between datasets, particularly between different species, are not as accurate[8,10–15]. These results suggest that 1) the features used to develop these algorithms are not effectively capturing changes in genomic context and 2) these features are influenced by factors other than DNA unwinding and/or cleavage at the target site, as illustrated in Fig. 1a. Broadly defined, "context"

includes all variables that can impact Cas9 activity independent of the biochemical cleavage reaction at the target site, while "genomic context" specifically refers to the host genomic factors excluding the target site, which have recently been shown to impact the rate at which the Cas9/gRNA complex finds its target[16]. Understanding how a gRNA's sequence influences Cas9/gRNA complex activity in different contexts may lead to improved algorithms and/or better predictions between species, as well as new avenues for engineering improved Cas9 variants.

A gRNA's sequence has been shown to influence Cas9/gRNA complex activity in a variety of ways (Fig. 1a). Secondary structure involving the targeting portion ("spacer") of the gRNA has been linked to low Cas9/gRNA complex activity due to decreasing the functional gRNA available[17,18]. Similarly, sequence features controlling expression of the gRNA have been linked to low activity in certain contexts. One example is four contiguous thymines in a row, which is a transcriptional pause signal when expressing gRNA from U6 promoters (Supplementary Fig. 1)[19]. Another example is that gRNA expressed from U6 promoters, in

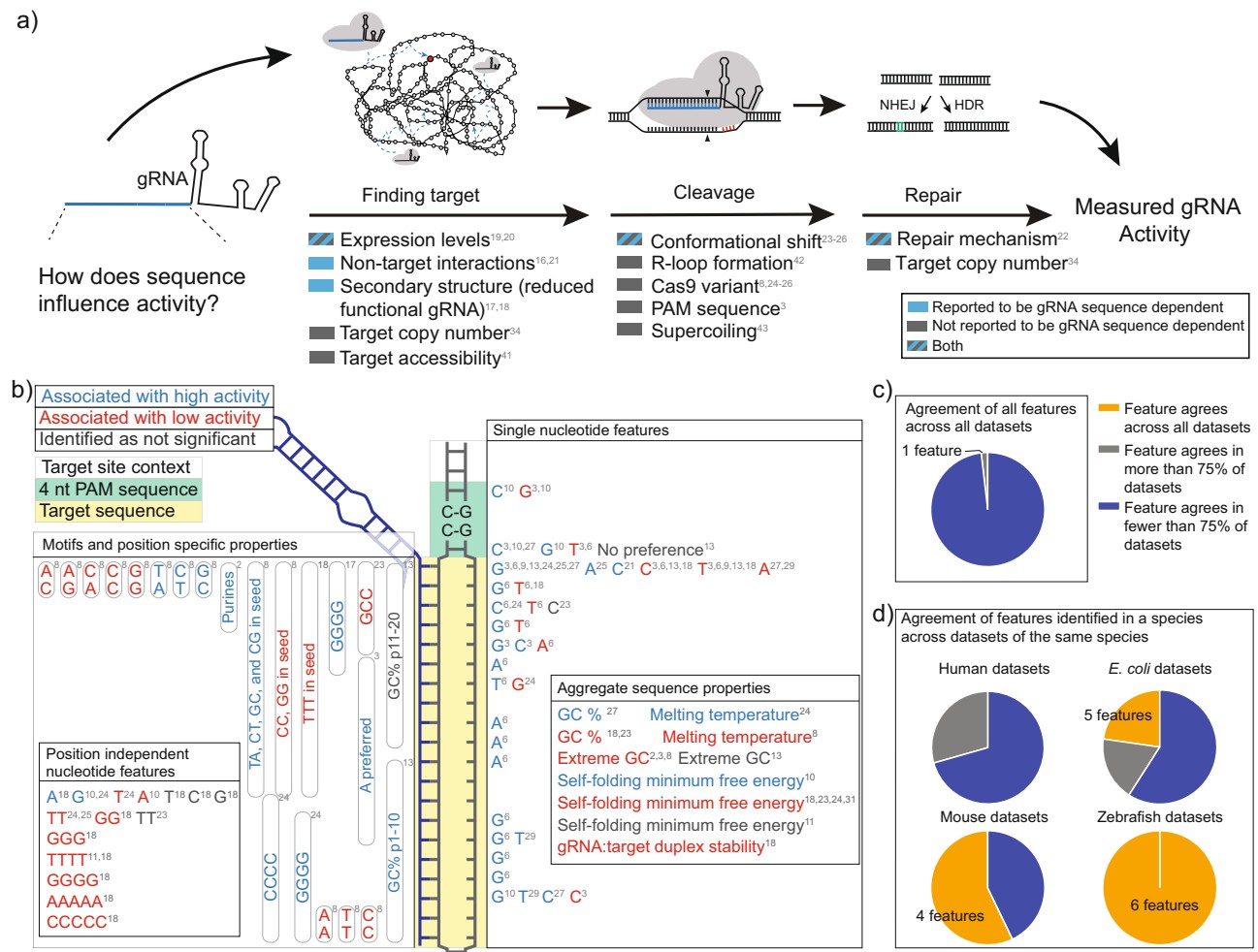

**Fig. 1 Summary of known factors that influence Cas9/gRNA complex activity and the reported gRNA sequence features identified as most important.** **a** The gRNA sequence can at least partially predict on-target activity. Here, we provide a model of factors reported to influence Cas9/gRNA complex activity and whether they impact finding the target site, cleaving the target, or repair of the target. **b** The gRNA:target duplex is highlighted in yellow and the PAM site is highlighted in green. Features that reportedly positively impact gRNA activity, have been identified as inhibitory to activity, or were specifically identified as not significant are labeled in blue, red, and gray, respectively. Position dependent features are labeled in the relevant position while position independent features are shown in separately labeled boxes. **c** When comparing all datasets, there was no feature identified with a common impact on activity across all datasets and only one feature identified across 75% of the datasets. **d** The consistency of reported features across datasets of the same species. Of features identified as important in human datasets, no feature consistently had the same impact across human datasets. For other species, we identified five, four, and 6 features that were consistent across datasets within the *E. coli*, mouse, and zebrafish datasets. *Y. lipolytica* was excluded from this analysis as there is only one dataset.

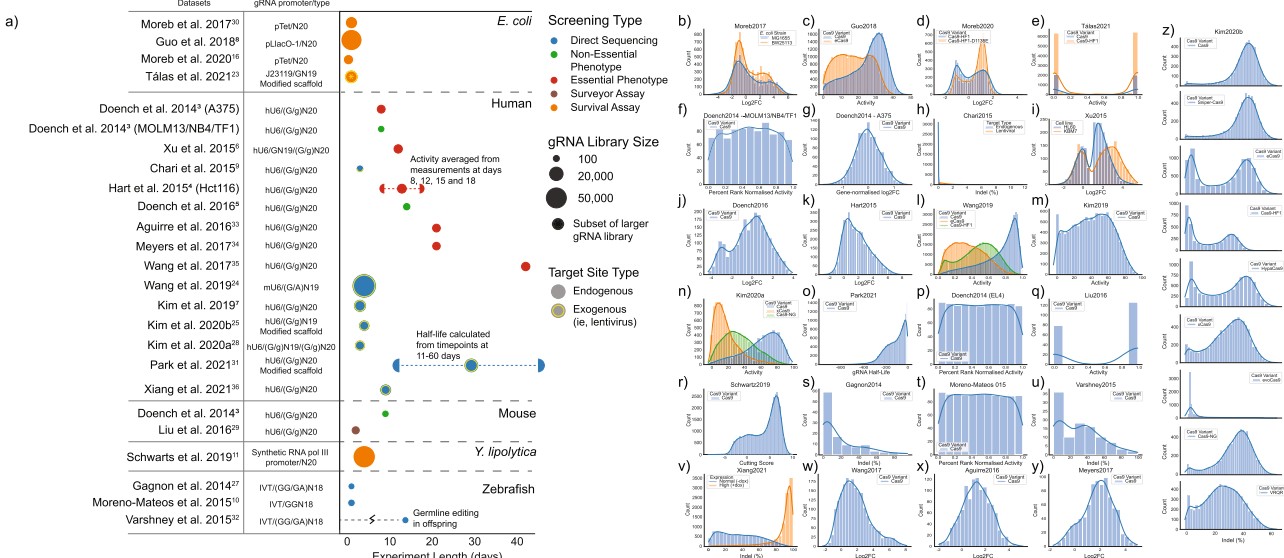

**Fig. 2 Summary of the datasets included in this analysis. a** Datasets collectively represent a diverse set of expression methods, experiment durations, species, selection types, gRNA library sizes, and target types. The bars on some data points represent the range of days that activity in those datasets has been calculated from. **b–z** These datasets cover a broad range of activity distributions, including moderate to extremely binary distributions, normal distributions, and completely uniform distributions, based on how activity was measured and data were processed. All datasets are shown with higher gRNA activity represented by larger numbers. In some cases, that required inverting the scale of activity as described in the Methods (namely, **b**, **d**, **o**, **w**, **x**, and **y**).

mammalian systems, need to start with a guanine for optimal expression[20]. The spacer sequence of a gRNA also determines the amount of time Cas9 spends interrogating non-target sites, which collectively make up the "search space" for a given Cas9/gRNA complex[16,21]. This leads to sequence dependent search times for different gRNA based on the non-target interactions throughout the genome[16]. Collectively, these sequence features influence how each Cas9/gRNA complex finds the target either by limiting expression of functional gRNA or slowing down search time (Fig. 1a). Cas9/gRNA complex activity has also been linked to gRNA sequence in other ways. For example, in eukaryotic systems, double strand break (DSB) repair primarily relies on non-homologous end joining (NHEJ). NHEJ is an error-prone repair process that has recently been shown to be predictable and largely dependent on the nucleotide 5' of the DSB, which would be the 4th nucleotide from the PAM site[22]. In some cases, this may influence the apparent or measured activity of a given gRNA. Finally, several studies have found sequence dependent differences in the activity of a given Cas9/gRNA complex when comparing wild-type Cas9 and higher fidelity variants that are known to cleave the target site more slowly[23–26]. This could indicate a role for sequence in determining the rate at which Cas9 cleaves the target site. Despite these links between sequence and activity, it is still not entirely clear which features contribute the most to gRNA specific activity and how these impact Cas9 activity in different genomic contexts.

Despite these varied and complex interactions, in the process of developing predictive algorithms for gRNA activity, many sequence-based features that predict activity have been identified (Fig. 1b). However, these features are generally not well explained mechanistically and are not consistent between different species or genomic contexts (Fig. 2c-d). In the present study, we sought to better understand the mechanism by which gRNA sequence impacts gRNA activity. Toward this goal, we report a retrospective analysis of 44 gRNA library datasets from different species, with several Cas9 variants, using both endogenous and exogenous target sites, various activity outputs, and different experimental systems[3–11,16,23–25,27–36]. In this analysis, we confirm species dependent differences in how sequence influences

gRNA activity and find strong evidence that gRNA sequence influences the time it takes for a given Cas9/gRNA complex to find the target site. This analysis sheds light on why gRNA prediction algorithms do not predict well between species and may lead to better predictions in the future.

## Results

**Compiled datasets represent different species, distributions of activity, and experimental methods**. We began by compiling data as discussed in the Methods Section, and illustrated in Fig. 2a[3–11,16,23–25,27–36]. Activity is reported as it was in the original dataset but we have inverted the sign on several datasets to ensure that in our comparisons more positive numbers correlate with more active gRNA. The datasets have varied distributions of cutting/cleavage activity, from binary distributions (gRNAs that either cut or do not cut) to skewed or normal distributions, suggesting significant experimental and context dependent differences in gRNA dependent activity (Fig. 2b, data compiled in Supplementary Data 1).

**Spacer independent PAM preference is consistent across species and genomic contexts**. To ensure that our analysis could identify sequence features that may impact the intrinsic Cas9 cleavage step (Fig. 1a), we first sought to confirm sequence features known to affect this activity. Cas9 has been reported to prefer NGGH PAM sites, where H is either A, C, or T[3]. We first determined how consistent this was across different contexts. To do this, we grouped gRNA within each dataset by the fourth nucleotide of the PAM, calculated average activity per group, and correlated these average values between datasets (Fig. 3a). As shown in Fig. 3b, we see strong correlations across almost all datasets, spanning different species (genomic contexts) and experiments. Furthermore, the datasets that do not correlate well have clear explanations. In the data from Chari et al. 2015, approximately 75% of the gRNA targeting endogenous loci are completely inactive, independent of the fourth nucleotide of the PAM (Fig. 3c). Several of the Cas9 variants have altered PAM preferences for the fourth nucleotide of the PAM when targeting NGG sites, which this analysis supports. Lastly, of

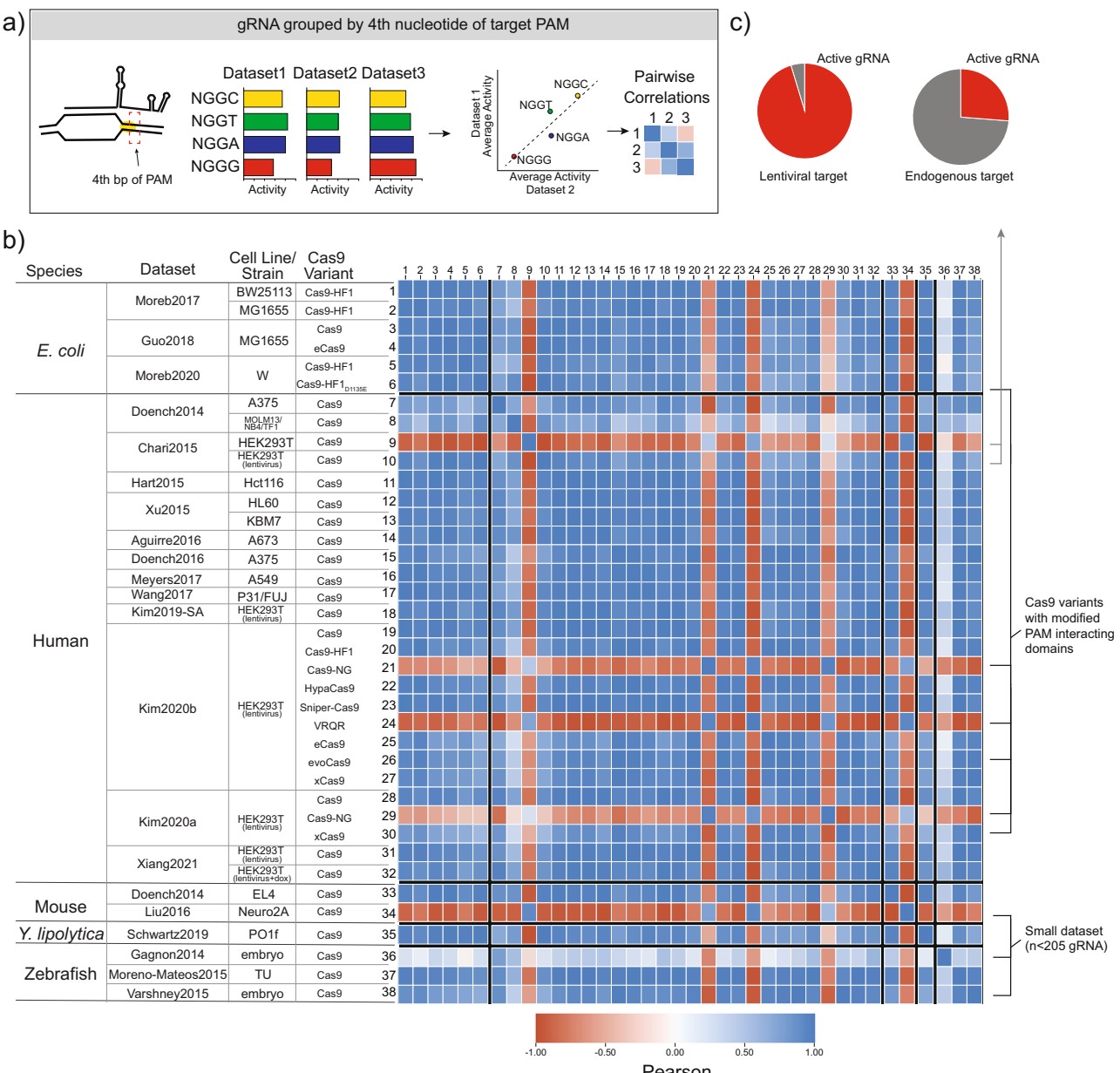

**Fig. 3 Validation of Cas9 PAM preference as an intrinsic feature of Cas9 activity. a** Cas9 has been reported to have a slight preference for NGGH PAM sites, where H is either A, C, or T. We grouped gRNA by the 4th nucleotide of the PAM site, calculated the average activity per group, and correlated the averaged values between datasets in a pairwise fashion. Pairwise correlations between all datasets show strong agreement in relative activity, confirming the preference for NGGH. Exceptions to this are well-explained and include datasets using Cas9 variants with modified PAM preferences or particularly small datasets which may not have the statistical power to capture known features. **c** Additionally, in Chari et al. 2015, most of the gRNA targeting endogenous sites showed no activity, overriding any impact that PAM preference may have. Among all datasets, three datasets did not have sequence variability in the 4th nucleotide of the PAM and were therefore excluded from this analysis: Wang et al. 2019, Tálas et al. 2021, and Park et al. 2021.

the three datasets targeting fewer than ~200 gRNA, two datasets are poorly correlated suggesting that small datasets may lack statistical power to capture even known features. While the preference for NGGH PAM sites is not new or novel, this analysis shows that even features that weakly influence activity but are intrinsic to Cas9 should be detectable across different experiments and species.

**The PAM proximal sequence is most predictive of Cas9/gRNA complex activity.** After confirming that known sequence features impacting intrinsic Cas9 activity can be quantified in these data, we next turned to evaluate which gRNA sequence features predict

Cas9/gRNA complex activity. It has been reported that the nucleotides in the PAM proximal region of the gRNA sequence (the "seed") are the most important features in predictive algorithms (Fig. 1)[3,6,8,9]. However, given that algorithms do not predict well between species, we sought to better understand which gRNA sequence features are most predictive of activity in different genomic contexts. gRNA sequence features are routinely digitized for these types of analyses using one hot encoding of overlapping dinucleotides[5]. We therefore used the same approach to digitize all sequences (Fig. 4a). For each dataset, after converting gRNA sequences to a one hot matrix encoding dinucleotides, we randomly split the dataset into a training group and testing group

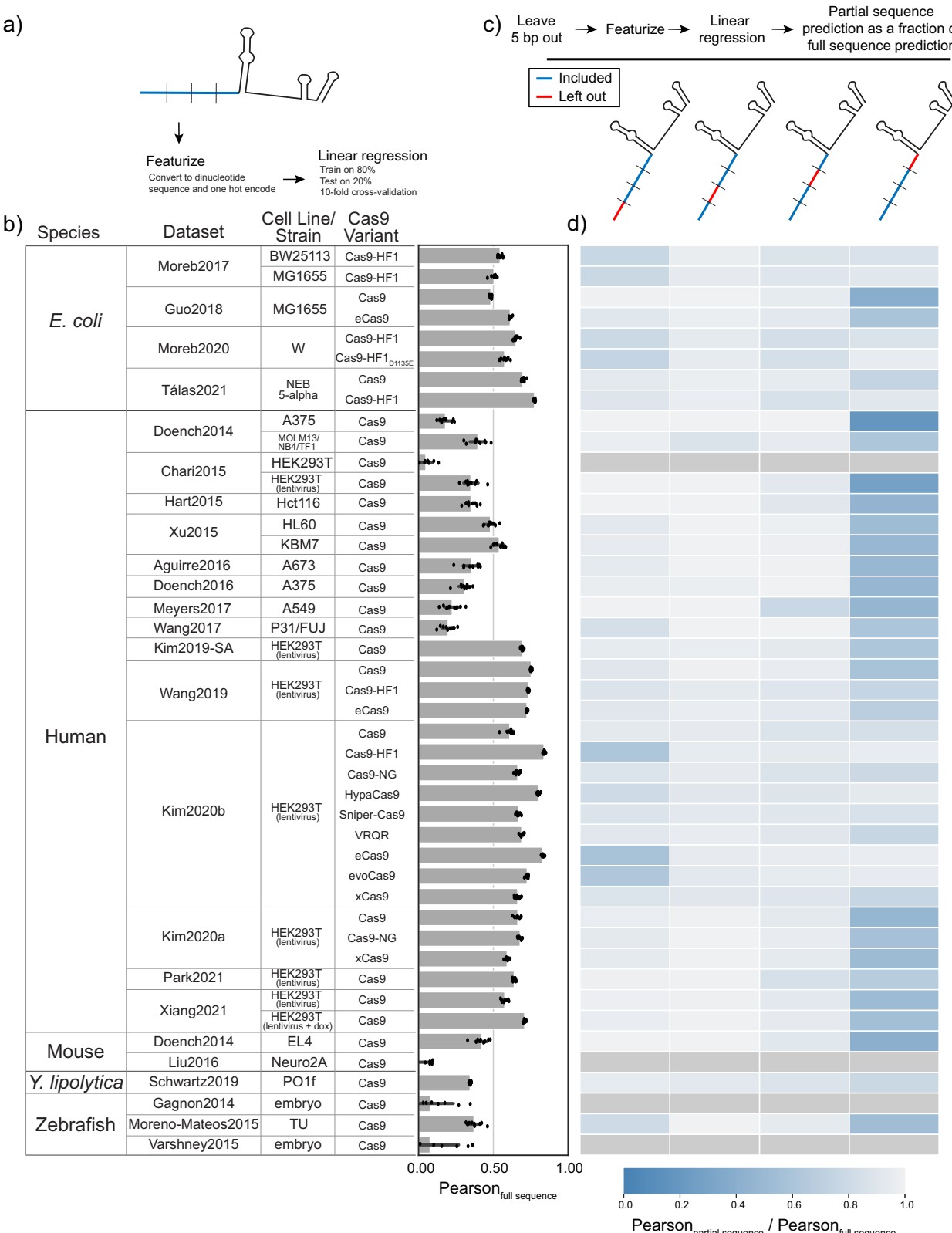

**Fig. 4 The PAM proximal portion of the gRNA provides most of the predictive power of the sequence. a** The 20 bp targeting sequence of each gRNA was converted to a dinucleotide one hot matrix and used to predict activity with a linear regression. For each dataset, 80% of gRNA were randomly assigned to a training group while the remaining 20% were used as a test group. Predicted activity was compared with actual activity using Pearson correlation coefficient and this process was repeated 10 times to achieve 10-fold cross validation. **b** The average of the 10-fold cross validation is shown for each dataset (along with individual results, dots). Error bars show the standard deviation of each cross validation. **c** We then repeated this analysis but left out 5 bp at a time. **d** The heatmap shows the averaged Pearson with 5 bp left out (Pearson$_{\text{partial sequence}}$) as a fraction of the averaged Pearson using all 20 bp (Pearson$_{\text{full sequence}}$). In cases where the Pearson$_{\text{full sequence}}$ average was close to zero, we excluded these datasets from further analysis.

representing 80% and 20% of the gRNA, respectively. After training, we predicted the activity in the test group and compared the predicted activity to actual activity using a Pearson correlation. We performed 10-fold cross validation by splitting the training and test groups randomly each time and averaging the results (Fig. 4b). As expected, and discussed above, for small datasets with n < 205 gRNA, this approach did not prove to be predictive. Similarly, in the data from Chari et al. 2015, the activity for gRNA targeting endogenous sites was not predictable, likely due to the low overall activity within this dataset (Fig. 3c). Among the remaining datasets, the Pearson values ranged from 0.18 to 0.84 highlighting both the link between sequence and activity and the variability of sequence impact in different contexts.

We next proceeded to iteratively repeat the linear regressions, each time removing one quarter of the gRNA sequence and correlating the remaining sequence with activity (Fig. 4c). The Pearsons for the partial sequence predictions as a fraction of the Pearson for full sequence prediction are given in Fig. 4d. These results highlight the majority of the predictive ability of the full gRNA sequence is from the PAM proximal region. The PAM distal 5 bp is also important for some datasets, primarily those utilizing high-fidelity variants of Cas9. This result is consistent with the PAM proximal features identified in the literature (Fig. 1b). This result supports the role for one mechanism underlying sequence dependent gRNA activity across datasets.

**The impact of the PAM proximal sequence on Cas9/gRNA complex activity is species or genomic context dependent**. We next turned to determine how the impact of the PAM proximal sequence varies as a function of the genomic context. To do so we examined specific sequence preferences within each dataset and between species. If nucleotide preference is derived from intrinsic Cas9 activity at the target site, we would expect to see strong agreement on sequence preference between datasets, regardless of species, similar to what is observed when looking at PAM preferences (Fig. 3)[2]. Intraspecies correlation with a reduced correlation between species would suggest that differences are driven by the larger genomic context in which Cas9 is deployed, including different inhibitory non-target site pools or other features that impact Cas9/gRNA complex search times[16]. The lack of any intra or interspecies correlation would suggest other confounding and unknown context dependent factors.

To investigate the species specific sequence preference in the PAM proximal position, we first determined what length of sequence to compare. In each dataset, we first measured the fractional representation of all possible k-mers (length 1 to 10) starting at the PAM proximal position (Fig. 5a). With the exception of the dataset from Hart et al. 2015, all datasets contained gRNA representing all 16 possible dinucleotide sequences in the PAM proximal position. In Hart et al. 2015, the dataset was designed to exclude thymines in the four PAM proximal positions, explaining the lack of specific dinucleotide sequences in this dataset[4]. Additionally, gRNA from libraries in Doench et al. 2014 and Doench et al. 2016 were included in the library design in Kim et al. 2019 (Supplementary Fig. 2), we therefore excluded those particular gRNA from the Kim et al. 2019 dataset to avoid redundancy. Similarly, some of the gRNA in Moreb et al. 2017 were also included in Moreb et al. 2020 and were removed from the Moreb et al. 2017 datasets for the purpose of this comparison. Refer to Supplementary Fig. 2 for a measure of gRNA redundancy among datasets. We grouped gRNA within the remaining datasets by the PAM proximal dinucleotide sequence, calculated the average activity for each group, and then looked at the correlation between these dinucleotide group

averages and activity between datasets in a pairwise-fashion, the results of which are given in Fig. 5b (see Supplementary Fig. 3 for grouped averages per dataset).

In these results, we see low interspecies correlations, but strong intraspecies correlations within the *E. coli*, human, and zebrafish datasets (Fig. 5c). In *E. coli*, there are strong correlations between our two previously reported datasets and that of Guo et al. 2018 but weak to no correlation with the datasets from Tálas et al. 2021. This study used an experimental design, including extrachromosomal targets, enabling rapid target cleavage. As a result, in Tálas et al. 2021, gRNA are mostly inhibited by the formation of unwanted secondary structures that render gRNA unable to bind the target site (the authors note predicted minimum free energy of gRNA secondary structure is strongly correlated with their library)[17,18,23]. Within the two mouse datasets, we don't see a good correlation but this is consistent with earlier results suggesting that the data reported by Liu et al. 2016 is not large enough and does not have high enough resolution to capture key sequence features driving activity. In *Y. lipolytica*, with only one dataset, we can only conclude that this dataset is not strongly correlated with other species, which agrees with the authors findings that several previously published predictive algorithms for both human and *E. coli* gRNA had no predictive ability on their dataset[11]. Similarly, within the three zebrafish datasets there is strong correlation when comparing data from Moreno-Mateos et al. 2015 with the other two but no correlation between the smaller datasets. Taken together, low interspecies correlations, but strong intraspecies correlations, is consistent with a model where species dependent differences in Cas9/gRNA complex activities are (1) a function of genomic context and (2) independent of intrinsic Cas9 cleavage activity[16].

This model is further supported by the fact that we see strong intraspecies correlations despite differences in the measurement of activity, and possible confounding variables such as sequence preferences for NHEJ repair[22]. For example, we see strong intraspecies correlations in experiments where either direct sequencing methods or phenotypic screening methods were used to measure activity. These results indicate that the method of measuring activity does not broadly alter gRNA sequence preference within a given genomic context. We further confirmed this by comparing the activity of gRNA measured in phenotypic screens in Doench et al. 2016 to the activity of the same gRNA measured by direct sequencing in Kim et al. 2019 (Supplementary Fig. 4). While there are differences in activity for individual gRNA, the averages based on the PAM proximal 2 bp are highly correlated. Another way to determine how the specific NHEJ repair outcomes may influence measured gRNA activity is through the use of inDelphi, a tool developed to predict NHEJ repair outcomes[22]. With inDelphi, it is possible to calculate the predicted frequency of frameshift mutations induced at specific target site. We therefore calculated an adjusted activity score based on this and found strong correlation between the adjusted activity and actual activity as well as strong correlation between the PAM proximal sequence preferences (Supplementary Fig. 4). We can conclude that the method of measuring activity is not a strong determinant of the sequence preference in the PAM proximal position.

While we have shown that the PAM proximal nucleotide preference is not influenced by the method of measuring activity, this does not mean other factors don't also influence activity in a given species or experiment. For example, in Fig. 4, gRNA sequence is less predictive of activity in the datasets targeting endogenous loci than in those targeting lentiviral target sites. This indicates either error in the measurement of activity in those datasets or a biological reason why sequence is less predictive of activity (ie, target accessibility or other confounding factors). So while the PAM proximal sequence impacts activity in a genomic

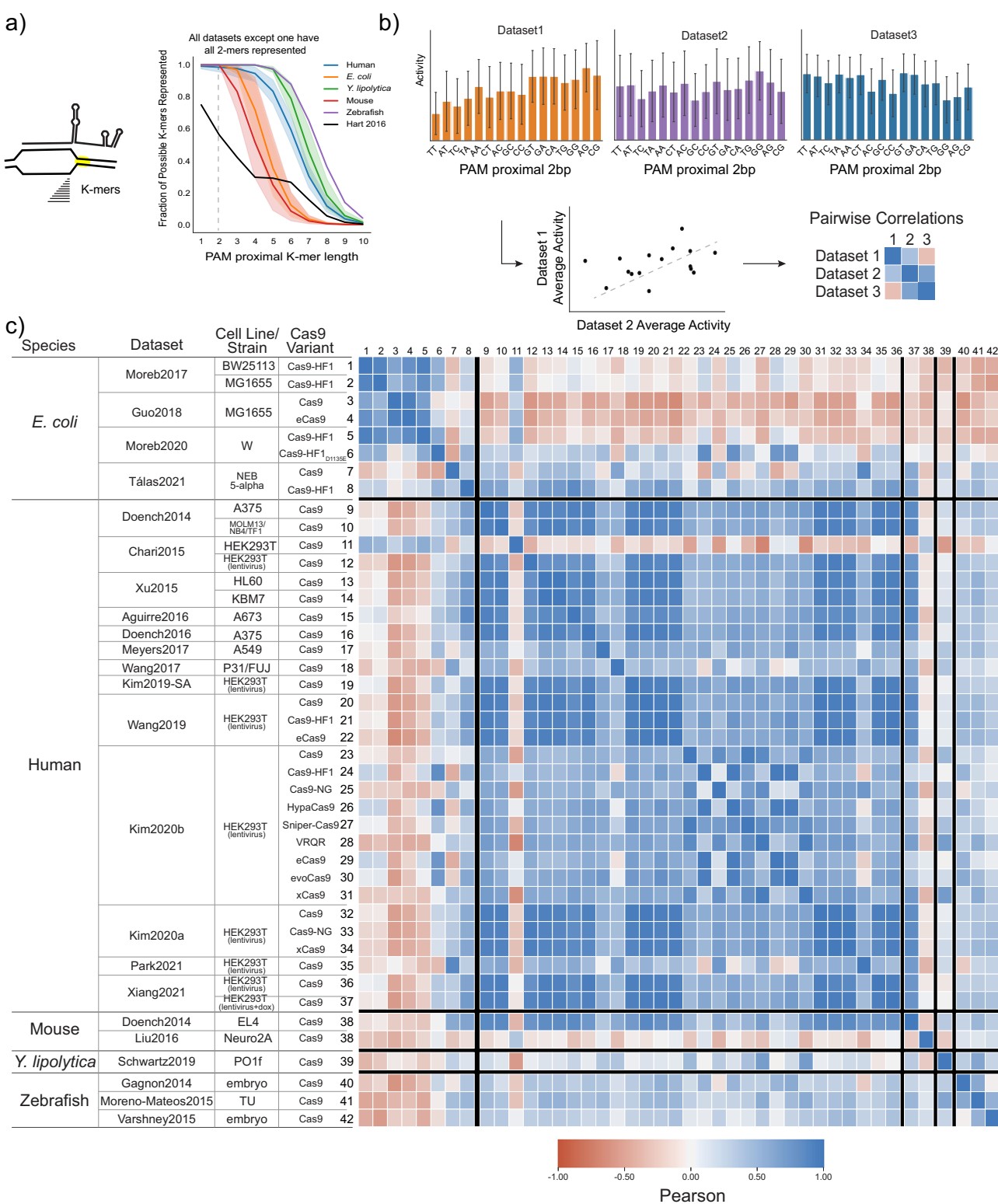

context dependent manner, this does not mean other factors do not influence or confound activity.

**The impact of the PAM proximal sequence on Cas9/gRNA complex activity can be attributed to the search space**. Within a given species, several factors reduce the intraspecies correlations, suggesting a reduced impact of the genomic context on gRNA sequence-dependent activity. In *E. coli* datasets, the addition of the D1135E mutation[37] reduces the correlation with other *E. coli* datasets (in contrast to other high fidelity variants and wild-type Cas9, Supplementary Fig. 3).The D1135E mutation is known to improve the specificity for NGG PAMs, thereby reducing the interactions at non-canonical PAMs for all gRNA and effectively reducing the size of the search space[16,37]. We previously reported

**Fig. 5 PAM proximal sequence preference is context dependent. a** We looked at the fraction of possible k-mers for each length, k, starting in the PAM proximal position. Lines and shaded regions represent the mean and one standard deviation from the mean, respectively, for each species. This shows that 2-mers are represented in all datasets, except Hart et al. 2015 which excluded thymines from the PAM proximal 4 bases. We therefore excluded Hart et al. 2015 from this analysis. We also excluded replicates of redundant gRNA present in another dataset: gRNA from Doench et al. 2014 and Doench et al. 2016 were removed from Kim et al. 2019 and gRNA from Moreb et al. 2020 were removed from Moreb et al. 2017 (see Supplementary Fig. 2 for more information). **b** We next grouped gRNA within each of the remaining datasets by the PAM proximal dinucleotide and calculated the average activity for each dinucleotide group (an example analysis is illustrated for datasets 1, 2, and 3). The averaged values for each data set were then correlated in a pairwise fashion between datasets to determine the similarity of dinucleotide sequence impact at this position. **c** The heatmap shows Pearson correlations between the averaged values for PAM proximal dinucleotides in all datasets, with blue being more positively correlated and red being more negatively correlated. Datasets are grouped by species and then ordered by year of publication. See Supplementary Fig. 3 for comparison of individual datasets.

that this resulted in higher overall on-target activity[16]. The reduced correlations with other *E. coli* datasets further suggests that this mutant also reduces the impact of genomic context on the PAM proximal sequence preference. There are also weaker correlations of the Kim et al. 2020b datasets and Park et al. 2021 dataset with other human datasets. Both of these datasets utilize different sgRNA scaffolds that modify the four consecutive thymines present towards the 5' end of the scaffold[25,31]. As four thymines in a row represent a known transcription terminator for the eukaryotic RNA polymerase III, the result of this modification is higher expression of the gRNA[38]. This suggests that increased gRNA expression reduces the impact of context on activity. The higher the number of Cas9/gRNA complexes in the cell, the faster these complexes can collectively find a given target (Fig. 1a). Lastly, both Wang et al. 2017[35] and Meyers et al. 2017[34] show modest but reduced correlations with other human datasets. Both of these datasets were algorithmically designed for improved activity which would be expected to reduce the sequence dependent difference between gRNA. Therefore, while algorithms may reduce sequence dependent differences between gRNA, both high gRNA expression and more specific PAM preference also reduce the influence of the PAM proximal sequence on gRNA activity (Supplementary Fig. 3). This result is consistent with reducing the time it takes a Cas9/gRNA complex to find the target.

**The impact of the PAM proximal sequence on Cas9/gRNA complex activity is independent of Cas9 nuclease activity.** Our analysis thus far supports the model, depicted in Fig. 1a, that gRNA sequence is primarily influencing the time it takes for a Cas9/gRNA complex to find the target site, rather than impacting intrinsic Cas9 cleavage activity or repair. While the datasets discussed above are of wild-type Cas9 or Cas9 variants with nuclease activity, we would expect to see similar sequence preference in nuclease null Cas9 variants, including deactivated Cas9 (dCas9) and base editors, if our hypothesis is correct. Using datasets from Horlbeck et al. 2016[39] (dCas9) and Marquart et al. 2020[40] (base editors) in human cells, we followed the same analysis of the PAM proximal 2 bp sequence that we did for other Cas9 datasets. We found that these datasets shared PAM proximal sequence preference with nuclease active Cas9 datasets in humans but had weaker or negative correlations in other species, despite not inducing double strand breaks (Fig. 6). This further supports the model that PAM proximal sequence influences the time it takes Cas9 to find the target site and is therefore independent of Cas9 nuclease activity.

**For a given genomic context, the PAM proximal sequence defines an upper limit of gRNA activity.** From our analysis, we conclude that gRNA sequence influences Cas9 search times in a genomic context-dependent manner. While we know that many factors can influence Cas9 activity (Fig. 1a), one implication of this is that the PAM proximal gRNA sequence should correlate

with an upper potential activity of a given gRNA. This is because the rate of finding the target site sets a fundamental upper limit on the activity of a given Cas9/gRNA complex. Additional factors such as target accessibility, supercoiling, and differences in repair (as well as others) would all negatively impact activity relative to this upper limit[22,41–43]. Another implication of gRNA sequence influencing Cas9 search times is that the sequence influences activity as a series of contiguous nucleotides, rather than a disparate set of dinucleotides as is commonly used in predictive algorithms[5–8,24]. This is because sequence-dependent search times are likely due to sequence-dependent non-target interactions throughout the genome in which Cas9 probes non-target sites in a zipper-like fashion from the PAM proximal position[16,21]. We therefore sought to determine if the contiguous PAM proximal sequence correlates with an upper potential limit to gRNA activity.

We previously looked at the fractional representation of all k-mers (length 1 to 10) in the PAM proximal position (Fig. 5a). From this analysis, we selected the two largest human datasets (from Wang et al. 2019 and Kim et al. 2019) that used the conventional gRNA scaffold in order to have full representation of all possible 5-mer sequences (Fig. 7a). We combined these datasets, grouped gRNA by their contiguous PAM proximal 5 bp sequence, calculated the average activity for each group and then used this averaged activity to predict gRNA activity in all human datasets based on the PAM proximal 5 bp for each gRNA (Fig. 7b). Unlike current prediction algorithms which treat gRNA sequences as a series of dinucleotides, this approach treats the PAM proximal sequence as one contiguous sequence. This better represents the manner in which the gRNA sequence sequentially binds to target and non-target sequences[16,21]. Upon correlating predicted activity with actual activity, we found that this approach was reasonably predictive for datasets using the conventional gRNA scaffold while less predictive of gRNA with a modified gRNA scaffold or higher expression levels, in line with our earlier analysis (Fig. 7c). We also confirmed that grouping by the PAM proximal 5 bases was more predictive than grouping by fewer nucleotides (Supplementary Fig. 4). Notably, with the exception of high fidelity variants and datasets with high expression levels, these predictions are comparable or better than our earlier linear regression-based predictions using the full gRNA sequence represented as dinucleotides. This suggests that using the complete sequence rather than a series of dinucleotides may better capture this aspect of gRNA activity. When looking at the residuals of these predictions, we noted a skewed distribution showing that this method often generates false positives but rarely generates false negatives (Fig. 7d–f). To better understand this, we plotted predicted activity against actual activity and, as expected, found that the PAM proximal sequence defines an upper bound of the potential activity for a given gRNA (Fig. 7g, h).

We then applied the same prediction to the nuclease null Cas9 datasets used earlier. The PAM proximal sequence is less able to predict activity in the dCas9 or base editing datasets (Fig. 7i).

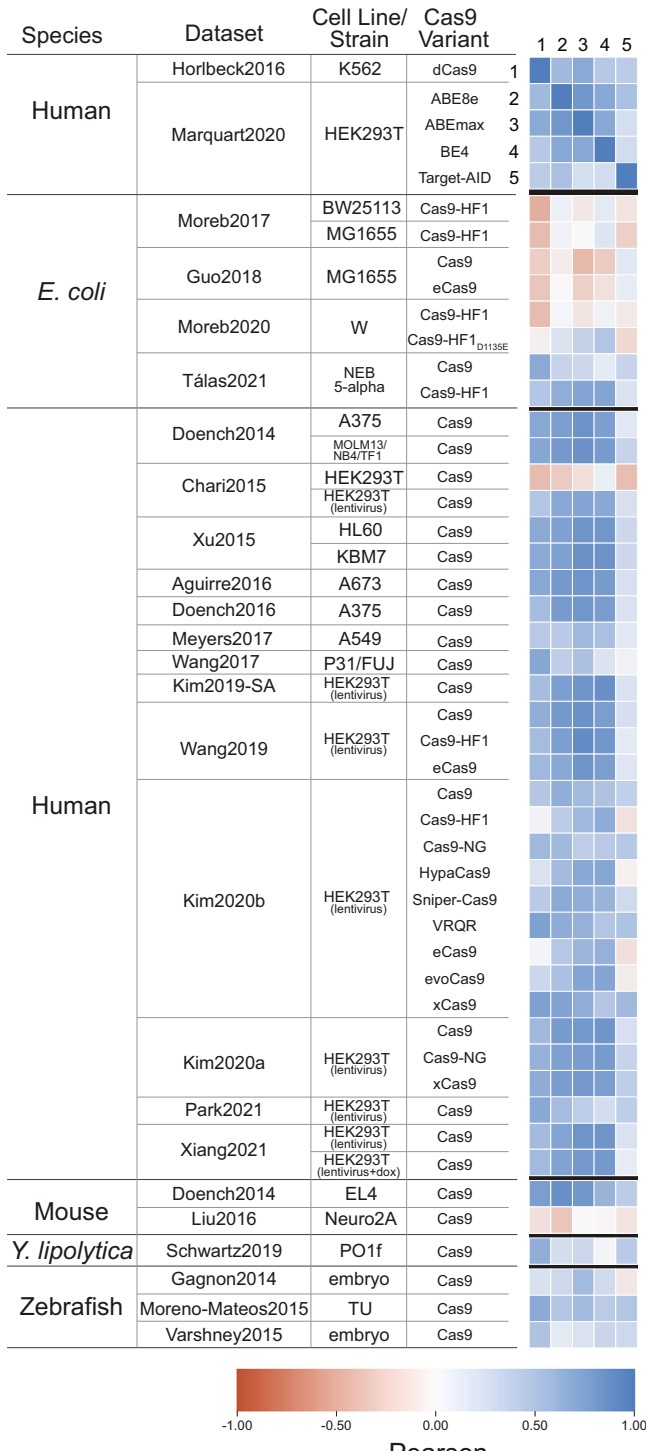

**Fig. 6 Nuclease null Cas9 variants have similar sequence preferences in the PAM proximal 2 bp.** Using a dCas9 dataset from Horlbeck et al. 2016[39] and base editing datasets from Marquart et al. 2020[40], we grouped gRNA by the PAM proximal 2 bp, averaged activity and correlated the grouped averages with the datasets from Fig. 5.

Both dCas9 and base editing have additional requirements to achieve high activity, such as correct positioning relative to the gene being inhibited for dCas9 or containing the correct sequence in the editing window for base editors, compared to wild-type Cas9[39,40]. This would be expected to lead to generally lower activity that is independent of gRNA sequence. Despite this, we found that the residuals were even more skewed than in Cas9 datasets (Fig. 7j), and when we plot predicted activity against actual activity, we see that the PAM proximal sequence can again mark an upper bound of the potential activity of a given gRNA (Fig. 7k). Taken together, this analysis provides further evidence that Cas9 activity is limited by finding the target site in a gRNA sequence-dependent manner.

## Discussion

The analysis presented here supports a model in which gRNA sequence at least in part dictates activity based on factors distal to the target site, i.e. genomic context, and as such, is primarily involved in determining the rate at which a given Cas9/gRNA complex finds its target site. We have shown a species-dependence between the sequence in the PAM proximal position and activity of a given gRNA (Fig. 5), even when adjusting for NHEJ repair outcomes (Supplementary Fig. 4) or when looking at nuclease null Cas9 variants (Fig. 6). This is in direct contrast to the four nucleotide PAM preferences observed across all species (Fig. 3) and indicates that the PAM proximal sequence preference is not influenced by intrinsic Cas9 biochemistry (such as target cleavage) or repair. Consistent with this, we also showed that within a given genomic context, the PAM proximal sequence of a given gRNA marks an upper bound of potential activity, supporting the idea that in general activity can be limited by how fast a specific Cas9/gRNA complex finds its target (Fig. 7, Supplementary Fig. 6). Further supporting this model, we show that both increased gRNA expression and increased PAM specificity, which should both decrease searching time, reduce the sequence preference in the PAM proximal position (Fig. 8). Together, these data and analyses suggest that sequence in the PAM proximal position of the gRNA influences the time it takes for a given Cas9/gRNA complex to find the target site. This effect can be attributed to the search space or the pool of inhibitory non-target sequences which vary as a function of genomic context, although other contextual features may also be at play[16].

This analysis helps to explain variability in gRNA activity across species. As illustrated in Fig. 8, species specific algorithms that predict gRNA activity may be useful but predicting between species is not appropriate with current gRNA sequence-based features[8,10,11,13]. Previously these differences in activity have been attributed to different Cas9/gRNA expression methods, different mechanisms of repair, target site accessibility, or phenotypic screening versus more direct methods of measuring activity[8,10,14,15]. Our analysis suggests that while expression levels matters, promoter differences are less likely to be driving differences between species. Similarly, differences in repair or target site accessibility may be impactful but would not explain the differences we observe in the PAM proximal sequence preferences between species. Furthermore, while better algorithms, such as deep learning[7,44], may improve species-specific predictions, a better understanding of genomic context and the impact of the search space will be required to predict activity across species. Understanding the impact of novel contexts on gRNA sequence dependent activity is key to developing CRISPR-based applications in new organisms, where current datasets are not expected to be predictive (Fig. 8e). To aid these future efforts, we have provided a proposed workflow and key considerations when designing gRNA libraries to better develop gRNA design algorithms in new systems (Supplementary Note 1).

This analysis also highlights factors that mitigate the impact of the PAM proximal sequence on activity and helps to explain differences observed in many studies. In particular, high expression levels of Cas9 and/or gRNAs can reduce the impact of the genomic context on gRNA activity, improving on-target activity

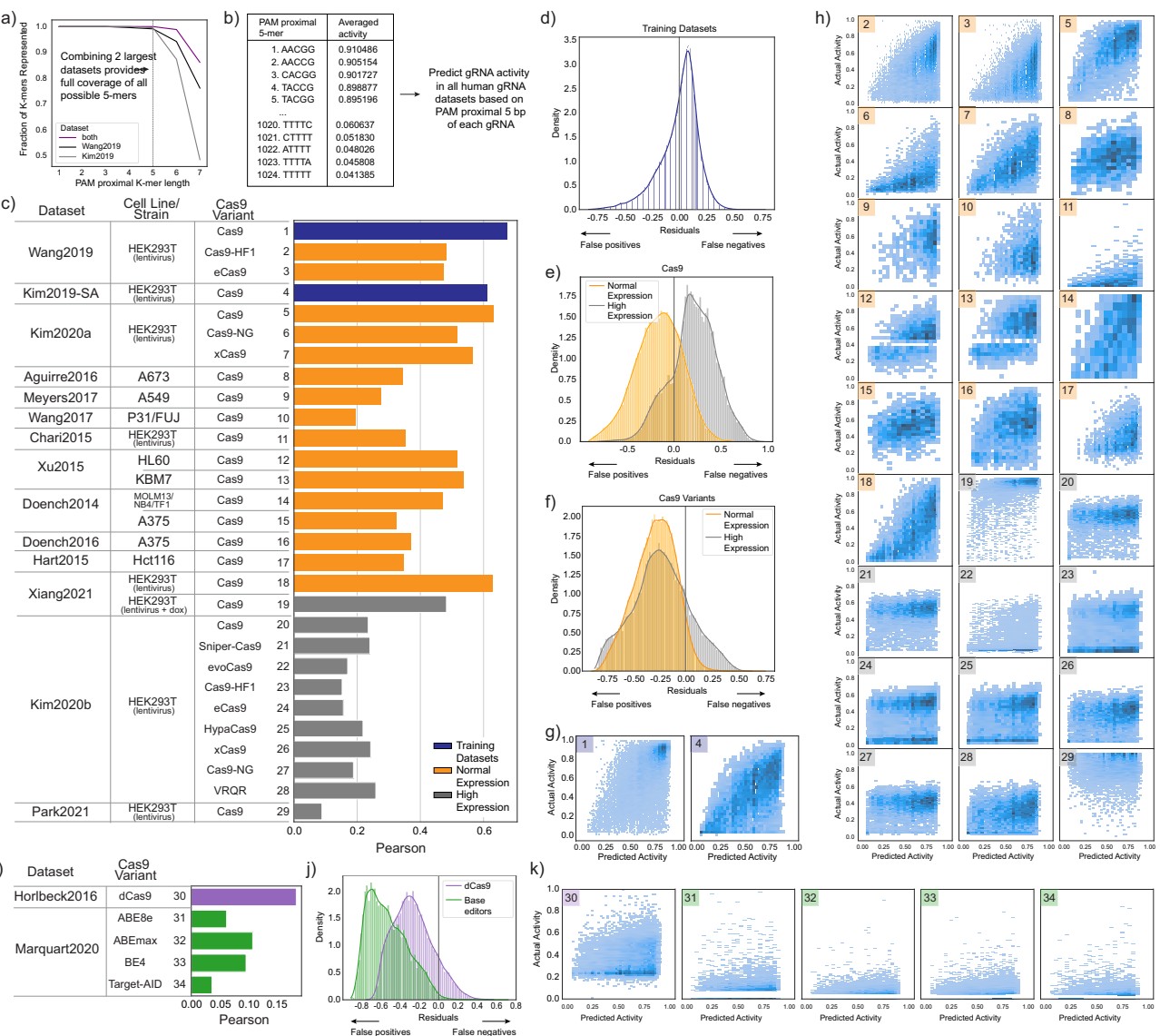

**Fig. 7 Within human datasets, the PAM proximal sequence defines an upper bound of potential activity for a given gRNA. a** Together, the two largest human datasets contain gRNAs that represent all possible 5-mers in the PAM proximal position. **b** We combined these datasets, grouped all gRNA by the PAM proximal 5 bp, calculated an average value for each group, and then used these grouped averages to predict gRNA activity in all human datasets. **c** We correlated this predicted activity with actual activity using Pearson correlation. The datasets that we used to generate the averages are highlighted in blue, while test datasets expressing gRNA normally or at higher expression levels are highlighted in orange and gray, respectively. We then calculated the residuals (Activity - Predicted Activity) for **d** the two training datasets, **e** all of the wild-type Cas9 datasets, and **f** all other Cas9 variants. Datasets with normal gRNA expression are in orange and those with higher gRNA expression are in gray. **g** For each of the training datasets we compared predicted activity on the x-axis to actual activity on the y-axis. **h** The same comparison is shown for each of the other datasets. Numbers in the top left of each sub-plot refer to the datasets identified in **c**). Refer to Supplementary Fig. 5 for a similar analysis for *E. coli* datasets. **i** We applied the same prediction method to a dCas9 dataset from Horlbeck et al. 2016[39] and base editing datasets from Marquart et al. 2020[40]. Shown are the Pearson correlations between Activity and Predicted Activity. We then **j** calculated the residuals and **k** plotted Predicted Activity against Actual Activity.

(Fig. 8d). However, this may not be a general solution as high expression levels are also correlated with increased off-target activity[45,46]. In cases where it is important to avoid off-target activity, other strategies may be preferred. One such strategy is to use Cas9 variants with higher PAM specificity (such as the D1135E mutant[37]), thus limiting the inhibitory non-target pool (Fig. 8c)[16,37,47]. Higher PAM specificity mutants may be particularly useful in host contexts where host specific predictive algorithms have not yet been developed.

In addition to strategies for improving Cas9 activity in different contexts, this analysis emphasizes that many factors may negatively influence Cas9 activity relative to a maximal activity

predicted by the PAM proximal sequence. While some of these factors, such as unwanted secondary structure in the gRNA or Cas9 preference for NGGH, are known, there is still much to learn[3,18]. For example, several reports have highlighted motifs or nucleotide preferences specific to high fidelity variants but mechanistic explanations for this are lacking[8,23–25]. Additionally, other contextual factors such as target site accessibility or other unknown chromosomal factors may play a role in Cas9 activity[7,8,41,48].

Understanding how the search space and genomic context more broadly impact on-target activity may also help elucidate factors impacting off-target activity. Our analysis supports that

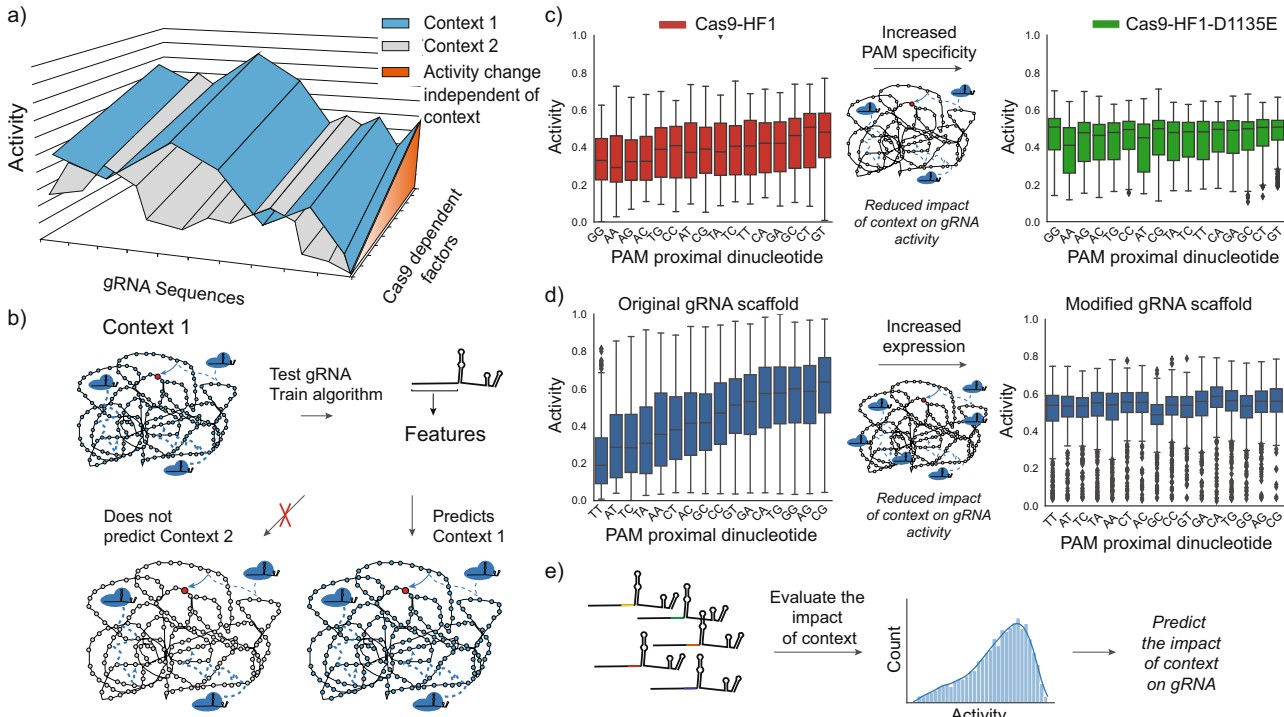

**Fig. 8 Cas9/gRNA complex activity is highly dependent on genomic context. a** A given gRNA can be expected to have different activity in different host organisms in a sequence specific manner. Cas9 dependent factors that are independent of host context present an orthogonal axis of activity. **b** Therefore, current algorithms trained on gRNA sequence features can perform well within the same context but will not accurately predict other species. However, the impact of context on gRNA activity can be reduced through **c** increasing the specificity of Cas9 PAM binding to reduce potential interactions at non-target sites (data from Moreb et al. 2020, each boxplot represents $n >= 209$ gRNA) and **d** increasing the expression of Cas9 and/or gRNA (data from Kim et al. 2019 and Kim et al. 2020b on left and right, respectively. Each boxplot represents $n >= 169$ gRNA). **e** Understanding the PAM proximal impact of context on gRNA activity also allows targeted gRNA libraries to specifically evaluate context effects (see Supplementary discussion on evaluating context).

in vivo on-target activity can be significantly influenced by factors outside of the target site. Since these contextual factors impact activity in a gRNA sequence-dependent manner, they are likely also relevant to off-target activity in the same way. Similarly, other unknown contextual factors may contribute to apparent sequence-dependence of off-target mismatch tolerance[45].

Finally, while much of the current understanding of Cas9 activity has been limited to a perspective focused on the target site, this analysis suggests that it may be equally important to understand sequence specific differences at interactions at non-target sites. It is our view that the sum of these transient interactions is a main driver of Cas9/gRNA complex search times. Several reports have found moderate or no connection between the number of predicted off-target sites and on-target activity[10,49]. However, off-target sites make up a small minority of the potential search space when including transient non-target interactions[16,21]. To our knowledge, transient interactions have only been evaluated in a handful of studies and no direct comparison of sequence dependent effects has been reported to date[16,21]. In vitro studies by Sternberg et al. demonstrated that Cas9 spent 1/10th the amount of time interrogating non-target sites with a 4 bp match than it did with non-target sites containing an 8 bp match[21]. However, for any given gRNA there are likely to be orders of magnitude more non-targets with 4 bp matches than with 8 bp matches, without accounting for possible mismatches, suggesting the sum of interactions at 4 bp matches would represent a larger search space than the sum of interactions at 8 bp matches. This suggests that understanding transient interactions may be crucial to developing a better understanding of the sequence features driving context dependent differences.

In the future, an improved understanding of gRNA sequence-specific Cas9 search times may well lead to (1) improved algorithms for predicting gRNA activity in established and novel organisms, (2) Cas9 variants with improved on-target and reduced off-target cleavage, (3) improved high-throughput functional screens, and (4) a better understanding of the factors driving activity in next generation CRISPR applications.

## Methods

**Compiling datasets**. We compiled 44 datasets from 23 papers (an overview is provided in Supplementary Data 1, while datasets grouped by species are provided in Supplementary Data 2-6)[3–11,16,23–25,27–36]. We first filtered the data, only including results for gRNA where (1) we could find a matching target site in the target genome (if targeting an endogenous site) and (2) gRNA were targeting NGG PAM sites. The following reference genomes were used: hg38 for human datasets (GenBank: GCA_000001405.15 [https://www.ncbi.nlm.nih.gov/assembly/GCF_000001405.15/])[50], mm9 for mouse datasets (GenBank: GCA_000001635.1 [https://www.ncbi.nlm.nih.gov/assembly/GCF_000001635.1])[51], danRer10 for zebrafish (GenBank: GCA_000002035.3 [https://www.ncbi.nlm.nih.gov/assembly/GCF_000002035.5/])[52], W29 for *Y. lipolytica* (GenBank: GCA_003054345.1 [https://www.ncbi.nlm.nih.gov/assembly/GCA_003054345.1])[53], MG1655 (GenBank: U00096.2 [https://www.ncbi.nlm.nih.gov/assembly/GCF_000005845.1/])[54], BW25113 (GenBank: CP009273.1 [https://www.ncbi.nlm.nih.gov/assembly/GCF_000750555.1/])[55] and W (GenBank: GCA_000184185.1 [https://www.ncbi.nlm.nih.gov/assembly/GCF_000184185.1/])[56] for *E. coli.*

We report activity as it was reported in the original dataset but have inverted the sign on several datasets to ensure that in our comparisons more positive numbers correlate with more active gRNA. Datasets for which we inverted the sign include Xu et al. 2015[6], Aguirre et al. 2016[33], Meyers et al. 2017[34], Wang et al. 2017[35], Moreb et al. 2017[30], Schwartz et al. 2019[11], Moreb et al. 2020[16], and Park et al. 2021[31]. The data in Supplementary Data 2-6 include this sign inversion. When plotting datasets together (as done in Figs. 4–7), we have re-scaled the activity measurements to values between 0 and 1, where 1 represents the most

active gRNA. This was done with min-max scaling from the Python library scikit-learn and the code can be found in Supplementary Software[57].

For several datasets, we only used a subset of the available data. From Hart et al. 2015[4], for example, we only used the data from the Hct116 cell line, as described by Haeussler et al. 2016[15]. This dataset included 4,239 gRNA with activity averaged over all time points provided from 8 to 18 days[4,15]. From Wang et al. 2014[2], we took data for cell lines KBM7 and HL60 that targeted essential genes, as provided by Xu et al. 2015[6]. For datasets from Aguirre et al. 2016[33], Meyers et al. 2017[34], and Wang et al. 2017[35], we chose one cell line and only selected gRNA targeting essential genes[58]. For datasets from Kim et al. 2020a[28], we only included gRNA Library B from the data provided at lentiviral MOI of 5 and only included gRNA targeting lentiviral sites. Similarly for Kim et al. 2020b[25], we only took data from gRNA Library B and we excluded repeat gRNA. From Park et al. 2021[31], we only took data from Library 1. Schwartz et al. 2019[11] performed library experiments in the presence and absence of the native NHEJ repair pathway. We used the Cutting Score results in the absence of NHEJ as this was not dependent on gene disruption by indels and therefore provided a more accurate measure of Cas9 activity[11]. In addition to the data collected in mouse and human cell lines in their lab, Doench et al. 2014[3] provide data extracted from Shalem et al. 2014[59] of gRNA targeting essential genes. From Tálas et al. 2021[23] we combined the "balanced" datasets provided by the authors as a subset of the larger ~1.2 million gRNA library. The "balanced" datasets were provided by the authors to better help differentiate features that drive differences in efficient and inefficient gRNA as the majority of the larger ~1.2 million gRNA library would be deemed efficient. Finally, from Xiang et al. 2021[36] we included both the dataset with dox added and the dataset without. For each dataset, we averaged activity for day 8 and day 10, similar to what the authors did.

For the nuclease null Cas9 variants, we collected dCas9 data from Horlbeck et al. 2016[39] and base editing data from Marquart et al. 2020[40] (Supplementary Data 7). For the dCas9 data, we used the data exactly as provided. For data from Marquart et al., they reported editing frequencies at each position within the gRNA sequence. Since we are only interested in how active the gRNA is rather than the editing outcome, we summed all editing activity per gRNA and used that as the activity measurement.

**Assessing the importance of previously reported sequence features**. We collected features specifically mentioned in the main text of papers as we reasoned this represents the features the authors deemed most important for activity. For each feature, we determined if it was a discrete feature (ie, guanine in position 20 of the gRNA) or continuous feature (ie, GC content). To determine if a discrete feature positively or negatively impacted gRNA activity in a specific dataset, we calculated a log odds ratio based on the frequency of said feature in the most active third of gRNA versus frequency in the least active third of gRNA. If the log odds ratio was negative, the feature was said to negatively impact gRNA activity and if it was positive it would be described as positively impacting activity. For continuous features, we used a Pearson correlation with gRNA activity to determine if the relative impact of a feature was positive or negative based on the sign of the correlation. A feature would be considered to be in agreement across all datasets if the sign of the log odds ratio or Pearson agreed across all datasets, indicating a uniformly positive or negative impact on gRNA activity. Calculations are provided in the code in Supplementary Software. Data is compiled in Supplementary Data 8.

**Computational analyses**. All computation was performed in Python with standard libraries, including: Datasets were managed with Pandas[60], NumPy was used for calculations[61], Regex was used for finding gRNA sequences in reference genomes[62], Scipy was used for statistics[63], and scikit-learn was used for linear regressions[57]. Seaborn and Matplotlib were used for plotting[64,65]. Biopython was used for calculating melting temperatures[66]. Folding energies of gRNA were calculated using ViennaRNA RNAfold package[67]. All code is provided in a Jupyter Notebook in Supplementary Software.

**Reporting summary**. Further information on research design is available in the Nature Research Reporting Summary linked to this article.

## Data availability

All data collected and used in this manuscript are freely available and attached as Supplemental Data. The code written for all analysis is included as a Jupyter Notebook file in the Supplemental Software file. The following reference genomes were used: hg38 for human datasets (GenBank: GCA_000001405.15)[50], mm9 for mouse datasets (GenBank: GCA_000001635.1 [https://www.ncbi.nlm.nih.gov/assembly/GCF_000001635.18/])[51], danRer10 for zebrafish (GenBank: GCA_000002035.3 [https://www.ncbi.nlm.nih.gov/assembly/GCF_000002035.5/])[52], W29 for *Y. lipolytica* (GenBank: GCA_003054345.1 [https://www.ncbi.nlm.nih.gov/assembly/GCA_003054345.1/])[53], MG1655 (GenBank: U00096.2 [https://www.ncbi.nlm.nih.gov/assembly/GCF_000005845.1/])[54], BW25113 (GenBank: CP009273.1 [https://www.ncbi.nlm.nih.gov/assembly/GCF_000750555.1/])[55] and W (GenBank: GCA_000184185.1 [https://www.ncbi.nlm.nih.gov/assembly/GCF_000184185.1/])[56] for *E. coli*.

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

## Acknowledgements

We would like to acknowledge the following support: ONR YIP #12043956, and DOE EERE grant #EE0007563. We would also like to acknowledge support from Duke Innovation & Entrepreneurship Initiative.

## Author contributions

E.A. Moreb performed computational analyses. E.A. Moreb and M.D. Lynch designed analyses, analyzed results, wrote, revised, and edited the manuscript.

## Competing interests

M.D. Lynch has a financial interest in DMC Biotechnologies, Inc., M.D. Lynch and E.A. Moreb have a financial interest in Roke Biotechnologies, Inc.
