## [Peer Review File · Nature Communications]

Reviewers' Comments:

Reviewer #1:

Remarks to the Author:

General comments:

The manuscript by Moreb, E.A. and Lynch, M.D. performed a retrospective analysis of how PAM proximal sequences affect CRISPR gRNA activities and sought to provide an explanation for the poor-prediction outcome of CRISPR gRNA activity data sets from different species. The study collected 39 datasets measured in five species (bacteria, yeast, human, mouse, zebrafish). The study first provides a summary on sequences features affecting gRNA activities and the agreement of features among different studies. The study also provides a summary of the 39 datasets, stratified by activity measurement types, library (gRNA) size of from each study, target site types, and distribution of gRNA activities from each dataset. Although no new info is provided by these two parts, the summary provides a very nice overview of the features, types of data set, variations within data set.

One concept of the study that the authors are trying to introduce is "genomic context", which the authors use to explain the difference in gRNA activities within data set and between species (studies). According to the authors' definition, "genomic context is referred to variables/DNA sequences outside the gRNA-Cas9 complex/gRNA". However, there is a fundamentally overlapping between the "genomic context" and the gRNA. What the authors referring to as "genomic context" is actually the "protospacer" and the 4-nt PAM region. More specifically, the study is analyzing the effect of PAM proximal sequences on gRNA activities and using these to explain the variations observed between different species. The studies first stratified all the gRNAs from each dataset according to the 2-mer sequences proceeding the PAM and observed there is a preference of some dinucleotide motifs, which give predictive outcome across some data set. The studies also analyzed data sets with longer PAM proximal sequences (5-mer), which give better predictive outcome as compared to shorter PAM proximal sequences motifs. As there is a generally poor (lower) correlation for the preference of PAM proximal sequences between data sets from difference species, author concludes that species dependent difference in gRNA activities and the impact of "genomic context" on gRNA activities.

In general, although the study has performed a sound collection, integration, presentation and analyses of the 39 published CRISPR gRNA data sets, the study provides very little new findings or knowledge about the impact of sequences context on CRISPR gRNA activity. Although the study trying to use the concept of "genomic context", what is actually presented is just sequence motifs (features) within the gRNA protospacer affecting gRNA activity. The term of "genomic context" is an inappropriate way of presentation. These sequences motifs have been reported by many gRNA activity prediction tools already. Although the study shows that 5-mer PAM proximal motif can give better prediction outcome, this semi-predictive approach does not provide an advantage as compared the already available ML- and deep-learning tools.

More specific comments

1. The impact of CRISPR activity determination methods on prediction outcome between studies. The current study tried to provide an explanation to the poor prediction outcomes for datasets from different species (studies) through the analyses of PAM proximal sequences. However, as already nicely summarized by the authors (Figure 2), there are so many confounding factors that affect the gRNA activities between these data sets. The most important one is methods used to determine the gRNA activities. Others factors like the time of editing, expression level and way of delivery collectively play an important role on gRNA activity.

(1) Direct sequencing, which should be regarded as the more accurate methods in deterring CRISPR gRNA activities. This method measures directly the indels introduced to the double strand breaks introduced by CRISPR Cas9. Although direct sequencing could not capture large indels, most indels after repairing of the DSBs are with the length of 30-nt.

(2) Non-essential (EGFP knockout), essential phenotypes and survival assay rely on the introduction of indels that least to lost-of-functions (or frameshift indels). Thus, not all indels introduced by CRISPR gRNA are captured by this method. Most importantly, several studies (e.g.

Nature, 2018. doi:10.1038/s41586-018-0686-x, Mol Cell. 2019 Feb 21; 73(4): 699–713.e6. doi: 10.1016/j.molcel.2018.11.031) had found that sequences flanking the DSB sites greatly affect the indel outcomes. Thus, the gRNA activities captured by this method will not necessarily reflect the actual gRNA-mediated cleavage activity.

(3) Surveyor Assay, which is only suitable for estimate and lack sensitivity. Some surveyor assay could give very low efficiency for gRNAs that results in nearly 100% indels.

Thus, although it is plausible to include all these 39 datasets, many confounding factors, particularly the method used to define gRNA activity, are also introduced by the different studies. The poor interspecies prediction outcome can be caused by many more confounding factors from the methods and experimental conditions. The conclusion that “genomic context” explains the interspecies difference on gRNA activities difference cannot be drawn from the current analyses, unless all other confounding factors have been minimized and excluded.

2. Issues of data redundancy should be taken into account

(1) To evaluate the prediction outcome (Figure 3), the study has used the strategy of 10-fold cross-validation. Each validation was trained with random sampling of 80% dataset and tested in the remaining 20% dataset. Although random sampling has been a common strategy, one frequently ignored confounding issue is the similarity of gRNA within the dataset. The analyses should check and remove redundant data from highly similar gRNAs (differed by less than 3 nt) from the training and test dataset.

(2) This data redundancy issue is more critical when comparing the mean gRNA activities based on PAM proximal sequences between the different studies (Figure 4 and 5). Many of the gRNAs evaluated in one study are also included in another study. The author should analyze how many gRNAs are overlapped between the different datasets from the same species and removed overlapping gRNAs from the correlation analyses. Many of the good correlation for intraspecies comparison could be an artifact of comparing the same gRNAs measured in different studies.

3. Inadequate judgements of the gRNA sequence and “genomic context”.

It appears that the study defines the gRNA spacer sequence and the “genomic context” as two different strategies, which is apparently wrong.

In the text page 3 “All on-target prediction algorithms heavily rely on gRNA sequence to predict activity. We therefore used the same approach to digitize all sequences”. This statement and judgement are not correct. Currently features used for machine- and deep-learning based CRISPR gRNA activity prediction are based typically based on 30-mer feature set, including the protospacer, PAM and flanking sequences (e.g.

<https://ieeexplore.ieee.org/stamp/stamp.jsp?tp=&arnumber=8115130> and Journal of Molecular Cell Biology, Volume 12, Issue 11, November 2020, Pages 909–911,

<https://doi.org/10.1093/jmcb/mjz116>). Thus, what the currently study analyzed give no additive novelty to what have been performed already. The introduction of “genomic context” do not provide any advantage as compared to current feature sets in CRISPR gRNA activity prediction.

The findings of favor and unfavored dinucleotides motifs have already been found by several previous studies. All these features have been incorporated in most activity prediction tools (e.g. Nature Communications 10(1) DOI: 10.1038/s41467-019-12281-8) – results of the feature importance.)

4. Rational on gRNA with longer PAM proximal-mers will have better correction (prediction).

The impact of PAM proximal sequences, more correctly sequence the seed region of the protospacer, have higher impact on CRISPR gRNA activity. This feature is well-known for the CRISPR system, there is no surprise that gRNAs with longer PAM proximal sequences will give a better correction and prediction outcome, as the longer the PAM proximal sequences are, the gRNAs used in training (average) and prediction are more similar.

Reviewer #2:

Remarks to the Author:

The authors present results from a nicely designed analytical framework aiming at determining the impact of genomic context on gRNA on-target efficiency, when performing pooled genome-wide CRISPR-cas9 screens.

Several studies have been published so far, each proposing a predictive method for gRNA activity based on their sequence content and intrinsic topological features. To my knowledge, this is the first attempt of an investigation of the effects of genomic context, namely i.e. the genomic sequence outside the gRNAs on their efficiency.

Toward their aim, the authors have performed a retrospective analysis of published datasets from screening with CRISPR-cas9 model systems from different species.

They have first determined which part of the gRNA sequence is the most predictive of activity within the same dataset, via a robustly cross-validated strategy reporting that the most informative portion of the gRNA sequences based methods is the region close to the PAM region, consistently across different datasets.

Then, the authors have investigated the high-activity preferential features within the PAM proximal regions reporting high intra specie similarities but no consistency across different species. From these results the authors conclude that the observed differences are driven by larger genomic sequence or context, that are specie dependent.

Briefly, this is a nice piece of work describing a simple yet ingenious analytical framework demonstrating the importance of the genomic context in determining gRNA efficiency and possibly explaining why all the algorithm designed so fare to predict sgNRA efficiency from their sequence do not validate well on data coming from species that are different from those they have been trained on.

The manuscript is well written, the presented analyses are well conceived and results support authors' claim.

I have few minor points, which I would like to see addressed or discussed in the response letter, before further considering this manuscript for publication.

1 - although the computational framework determining the most informative portions of the gRNA sequences is well designed and robustly cross-validated, the authors just used linear regression as predictive method, not even testing a single method of those presented in the publications from which the datasets are derived. Wouldn't be worthy to reperform the analysis including more methods?

2 - several other (much larger datasets) of the same kind of those analysed on this manuscript are now on the public domain (for example those introduced in PMID: 29083409 and PMID: 30971826). Including these datasets in the analyses would be a nice plus

3 - the authors should briefly explain how gRNA activity is computed and how they account for gene targeted function? is the analysis restricted to 'essential genes' for example?

We thank the reviewers for the careful review and helpful comments. We have made specific changes to the manuscript as detailed below. More generally we appreciated that the core of the story was not coming through as intended. As a result we have made extensive changes to the manuscript, including moving some figures from Supplemental to the main text, clarifying language throughout and even changing the title for clarity.

REVIEWER COMMENTS

Reviewer #1 (Remarks to the Author):

General comments:

The manuscript by Moreb, E.A. and Lynch, M.D. performed a retrospective analysis of how PAM proximal sequences affect CRISPR gRNA activities and sought to provide an explanation for the poor-prediction outcome of CRISPR gRNA activity data sets from different species. The study collected 39 datasets measured in five species (bacteria, yeast, human, mouse, zebrafish). The study first provides a summary on sequences features affecting gRNA activities and the agreement of features among different studies. The study also provides a summary of the 39 datasets, stratified by activity measurement types, library (gRNA) size of from each study, target site types, and distribution of gRNA activities from each dataset. Although no new info is provided by these two parts, the summary provides a very nice overview of the features, types of data set, variations within data set.

One concept of the study that the authors are trying to introduce is “genomic context”, which the authors use to explain the difference in gRNA activities within data set and between species (studies). According to the authors’ definition, “genomic context is referred to variables/DNA sequences outside the gRNA-Cas9 complex/gRNA”. However, there is a fundamentally overlapping between the “genomic context” and the gRNA. What the authors referring to as “genomic context” is actually the “protospacer” and the 4-nt PAM region.

We apologize that our definition of genomic context was not clear. This misunderstanding seems to be at the heart of many of the reviewer’s concerns. We intended for “genomic context” to be understood as factors independent of the target site that may affect gRNA activity. This would exclude factors that impact DNA unwinding, cleavage, and/or repair but would include, for example, the pool of inhibitory non-target sites throughout a given genome (Moreb *et al* 2020. PMID: 33346713) or other factors that impact Cas9 finding the target site. We have updated our definition on page 2, line 42: “Broadly defined, “context” includes all variables that can impact Cas9 activity independent of the biochemical cleavage event at the target site while “genomic context” specifically refers to the host genomic DNA, excluding the target site, which may play an inhibitory role in Cas9 finding the target.” Additionally, to better make this point we have attempted to change our wording throughout the manuscript to highlight that we are showing that finding the target site (updated panel in Figure 1a, copied below) has significant sequence specific impacts on gRNA activity that changes between different species contexts. Sequence independent effects impacting finding the target

have been identified previously (ie, more expression = higher activity) as well as our own publication on sequence dependent effects in which Cas9 interacts at non-target sites in a sequence dependent manner (Moreb *et al* 2020. PMID: 33346713). Therefore, this analysis is the first to demonstrate that sequence specific gRNA activity is largely attributable to how Cas9 finds a specific target, rather than activity at the target or repair (indels are discussed in a later reply).

More specifically, the study is analyzing the effect of PAM proximal sequences on gRNA activities and using these to explain the variations observed between different species. The studies first stratified all the gRNAs from each dataset according to the 2-mer sequences proceeding the PAM and observed there is a preference of some dinucleotide motifs, which give predictive outcome across some data set. The studies also analyzed data sets with longer PAM proximal sequences (5-mer), which give better predictive outcome as compared to shorter PAM proximal sequences motifs. As there is a generally poor (lower) correlation for the preference of PAM proximal sequences between data sets from difference species, author concludes that species dependent difference in gRNA activities and the impact of “genomic context” on gRNA activities.

In general, although the study has performed a sound collection, integration, presentation and analyses of the 39 published CRISPR gRNA data sets, the study provides very little new findings or knowledge about the impact of sequences context on CRISPR gRNA activity. Although the study trying to use the concept of “genomic context”, what is actually presented is just sequence motifs (features) within the gRNA protospacer affecting gRNA activity. The term of “genomic context” is an inappropriate way of presentation. These sequences motifs have been reported by many gRNA activity prediction tools already. Although the study shows that 5-mer PAM proximal motif can give better prediction outcome, this semi-predictive approach does not provide an advantage as compared the already available ML- and deep-learning tools.

We appreciate the reviewer’s perspective, but emphasize that the main goal of this study is not to improve predictions of any given data set, but rather to understand why sequence can predict activity. There are many possible explanations for this, as the reviewer notes elsewhere, and we have highlighted these in Figure 1a. The novelty of this work is the demonstration of a new fundamental understanding that protospacer sequence influences Cas9 activity not due to differences in a given Cas9/gRNA complexes ability to cleave DNA, but rather its ability to find a

target site and that the rate of finding a given target site is dependent on genomic sequences (context or search space) outside of the protospacer and PAM. We apologize that this was not more clear and have significantly updated the paper to better represent this, including updating the title, re-wording the manuscript, and updating figures. We also included dCas9 and base editing datasets in Figure 6 and 7.

More specific comments

1. The impact of CRISPR activity determination methods on prediction outcome between studies.

The current study tried to provide an explanation to the poor prediction outcomes for datasets from different species (studies) through the analyses of PAM proximal sequences. However, as already nicely summarized by the authors (Figure 2), there are so many confounding factors that affect the gRNA activities between these data sets. The most important one is methods used to determine the gRNA activities. Others factors like the time of editing, expression level and way of delivery collectively play an important role on gRNA activity.

We thank the reviewer for bringing up confounding variables. We highlighted many of these in Figure 1 and 2 and also discussed them in the Discussion (Page 11, Line 320). In fact, these confounding variables make a stronger case for the impact of genomic context and the conclusions of our analysis. If these many confounding variables had a larger impact on Cas9/gRNA complex activity than the species dependent context, we would not see strong correlations between studies of the same species where these confounding variables are present (as seen in Fig 5). Despite these numerous confounding variables we still see very strong intraspecies correlations and low interspecies correlations, which points to the greater importance of context on activity when compared to these confounding factors (which are expected to decrease correlations between data sets of the same species).

(1) Direct sequencing, which should be regarded as the more accurate methods in deterring CRISPR gRNA activities. This method measures directly the indels introduced to the double strand breaks introduced by CRISPR Cas9. Although direct sequencing could not capture large indels, most indels after repairing of the DSBs are with the length of 30-nt.

(2) Non-essential (EGFP knockout), essential phenotypes and survival assay rely on the introduction of indels that least to lost-of-functions (or frameshift indels). Thus, not all indels introduced by CRISPR gRNA are captured by this method. Most importantly, several studies (e.g. Nature, 2018. doi:10.1038/s41586-018-0686-x, Mol Cell. 2019 Feb 21; 73(4): 699–713.e6. doi: 10.1016/j.molcel.2018.11.031) had found that sequences flanking the DSB sites greatly affect the indel outcomes. Thus, the gRNA activities captured by this method will not necessary reflecting the actual gRNA-mediated cleavage activity.

(3) Surveyor Assay, which is only suitable for estimate and lack sensitivity. Some surveyor assay could give very low efficiency for gRNAs that results in nearly 100% indels.

We thank the reviewer for bringing up these valid points (1-3). We also believe these to be important and have highlighted them in Figure 2 and agree on the relative order of accuracy

of these methods. However, in looking at gRNA activity grouped by the PAM proximal 2 bp, we don't see differences between direct sequencing and phenotypic screening within human datasets (Figure 5). This means that the accuracy of these measurements is not confounding the relationships we have identified. We have added a paragraph to discuss this point in our results (Page 7, line 186) as well as including including Supplemental Figure S4, copied below:

Figure S4: Assessing the impact of gRNA screening method on PAM proximal sequence-based averages. One of the challenges in assessing gRNA activity in phenotypic screens is that the indels generated do not always lead to gene knock-outs. Since indel formation is somewhat predictable and sequence dependent, we wanted to assess how differences in measuring gRNA activity might influence the PAM proximal sequence preference. 1,899 gRNA from Doench *et al.* 2016⁵ were included in the library screened by Kim *et al.* 2019⁶. The Doench *et al.* screen used a phenotypic readout of activity while Kim *et al.* used a more direct sequencing based readout. a) The measured activity for each gRNA is modestly correlated ($r=0.484$) between the two screens. b) However, if we group gRNA by the PAM proximal 2 base pairs and calculate the average activity per group, we see a strong correlation between these averaged values ($r=0.908$). This suggests that the relative impact of the PAM proximal sequence is the same within each dataset, despite the differences in how the screens measure activity. To further assess this, we used inDelphi to calculate an “inDelphi Adjusted Activity” score for the dataset from Hart *et al.* 2015.^{7,8} This dataset screen was performed in cell line Hct116, one of the cell lines included in the inDelphi model. inDelphi provides a predicted “Frameshift Frequency” for a given DNA break based on the surrounding nucleotides. Given that measured activity in a phenotypic screen would only capture frameshifts which lead to knock-outs, we reasoned that: Actual Activity \times Frameshift Frequency = Measured Activity. c) Using this logic, we calculated what the actual activity should be (“inDelphi Adjusted Activity”) and compared it to the measured activity for each gRNA. We found a strong correlation ($r=0.795$) between measured Activity and inDelphi Adjust Activity, suggesting that phenotypic screens are mostly capturing gRNA activity accurately. d) Furthermore, we found that when calculating the average activity by the PAM proximal 2bp, the averaged inDelphi Adjusted Activity was strongly correlated with

the averaged Measured Activity. This further confirms the strong correlations between PAM proximal sequence-based averages in human datasets that we see in Figure 4, despite differences in screening methods

In addition, one of the conclusions of our analysis is that a large part of the impact of sequence on activity is not directly related to cleavage or repair activity at the target site but rather is linked to the ability of a given Cas9/gRNA complex to find the target site. Further supporting this, we have included Figure 6 in which we identify the same species specific trends in nuclease-null Cas9 variants by looking at dCas9 and base editing variants. We also show that the PAM proximal sequence dictates an upper potential limit for a given gRNA for these same datasets, similar to what we see in nuclease active Cas9 (Figure 7).

Last, we acknowledge and agree with the reviewer about the relative merit of the surveyor assay in screens such as these. However, the only dataset in this collection that is based on surveyor assays has been used in previous studies of gRNA activity and therefore we felt it appropriate to include (Hauessler et al 2016, PMID: 27380939). We specifically call out bad correlations related to this dataset and multiple times highlight the low fidelity of this dataset.

Thus, although it is plausible to include all these 39 datasets, many cofounding factors, particularly the method used to define gRNA activity, are also introduced by the different studies. The poor interspecies prediction outcome can be caused by many more cofounding factors from the methods and experimental conditions. The conclusion that “genomic context” explains the interspecies difference on gRNA activities difference cannot be drawn from the current analyses, unless all other cofounding factors have been minimized and excluded.

As discussed above, we have to respectfully disagree with the reviewer’s comment, the cofounding variables discussed above are more prevalent within a given species and are a function of the number and diversity of data sets (There are more human data sets with more cofounding factors among them). The overarching agreement within species suggests that the cofounding variables are not negating the relationships we have identified. Furthermore, across all species the sequence context of the PAM site is consistent (Figure 3), suggesting that the influence of key consistent features is detectable across datasets despite the cofounding variables.

2. Issues of data redundancy should be taken into account

(1) To evaluate the prediction outcome (Figure 3), the study has used the strategy of 10-fold cross-validation. Each validation was trained with random sampling of 80% dataset and tested in the remaining 20% dataset. Although random sampling has been a common strategy, one frequently ignored cofounding issue the similarity of gRNA within the dataset. The analyses should check and remove redundant data from highly similar gRNAs (differed by less than 3 nt) from the training and test dataset.

(2) This data redundancy issue is more critical when comparing the mean gRNA activities based on PAM proximal sequences between the different studies (Figure 4 and 5). Many of the gRNAs evaluated in one study are also included in another study. The author should analyze how many gRNAs are overlapped between the different datasets from the same species and removed

overlapping gRNAs from the correlation analyses. Many of the good correlation for intraspecies comparison could be an artifact of comparing the same gRNAs measured in different studies.

We agree that redundancy would skew the results, and have assessed the level of redundancy in these 39 (now 44 data sets). This analysis is given in Supplemental Figure S2. Most datasets are largely unique however the gRNA from Doench *et al* 2014 and Doench *et al* 2016 were included in a later screen from Kim *et al* 2019. We therefore removed that set of gRNA from Kim *et al* 2019 for Figure 5. We also removed gRNA from Moreb *et al* 2017 that are present in Moreb *et al* 2020. We believe that this addresses any major concerns around data redundancy.

3. Inadequate judgements of the gRNA sequence and “genomic context”.

It appears that the study defines the gRNA spacer sequence and the “genomic context” as two different strategies, which is apparently wrong.

We are confused by this comment, which we think comes from the misunderstanding of our definition of “genomic context”. Hopefully, our clarifications above and major changes to the text have resolved this concern, although we are happy to address any new or clarified concerns.

In the text page 3 “All on-target prediction algorithms heavily rely on gRNA sequence to predict activity. We therefore used the same approach to digitize all sequences”. This statement and judgement are not correct. Currently features used for machine- and deep-learning based CRISPR gRNA activity prediction are based typically based on 30-mer feature set, including the protospacer, PAM and flanking sequences (e.g.

<https://ieeexplore.ieee.org/stamp/stamp.jsp?tp=&arnumber=8115130> and Journal of Molecular Cell Biology, Volume 12, Issue 11, November 2020, Pages 909–911,

<https://doi.org/10.1093/jmcb/mjz116>). Thus, what the currently study analyzed give no additive novelty to what have been performed already. The introduction of “genomic context” do not provide any advantage as compared to current feature sets in CRISPR gRNA activity prediction.

Again, we apologize for the confusion. While it’s true that current features are typically based on the 30-mer sequence, the spacer (or protospacer) sequence tends to contain the most predictive features (Figure 1), hence the words “heavily rely”. We have removed this sentence and updated this paragraph to be more clear (Page 4, line 111). Hopefully, our updated language and definition around “genomic context” has also been clarified sufficiently.

The findings of favor and disfavored dinucleotides motifs have already been found by several previous studies. All these features have been incorporated in most activity prediction tools (e.g. Nature Communications 10(1) DOI: 10.1038/s41467-019-12281-8) – results of the feature importance.)

We agree with this comment, and again highlight that we are not debating the importance of favored and unfavored dinucleotides in prediction of activity, but rather shedding light on the reason for this importance.

4. Rational on gRNA with longer PAM proximal-mers will have better correction (prediction). The impact of PAM proximal sequences, more correctly sequence the seed region of the protospacer, have higher impact on CRISPR gRNA activity. This feature is well-known for the CRISPR system, there is no surprise that gRNAs with longer PAM proximal sequences will give a better correction and prediction outcome, as the longer the PAM proximal sequences are, the gRNAs used in training (average) and prediction are more similar.

We agree with this comment, and again highlight that we are not debating the importance of sequence stretches in the prediction of activity, but rather shedding light on the reason for this importance. However, we would note that most predictive algorithms treat the gRNA sequence as a series of overlapping dinucleotides or independent single nucleotides. One implication of our analysis, and one reason this analysis is novel, is that we have treated the sequence as a contiguous five nucleotide sequence (Figure 7). This enabled us to look at the structure of the data in a new way and conclude that the PAM proximal sequence dictates the upper potential activity for a given gRNA.

Reviewer #2 (Remarks to the Author):

The authors present results from a nicely designed analytical framework aiming at determining the impact of genomic context on gRNA on-target efficiency, when performing pooled genome-wide CRISPR-cas9 screens.

Several studies have been published so far, each proposing a predictive method for gRNA activity based on their sequence content and intrinsic topological features. To my knowledge, this is the first attempt of an investigation of the effects of genomic context, namely i.e. the genomic sequence outside the gRNAs on their efficiency.

Toward their aim, the authors have performed a retrospective analysis of published datasets from screening with CRISPR-cas9 model systems from different species.

They have first determined which part of the gRNA sequence is the most predictive of activity within the same dataset, via a robustly cross-validated strategy reporting that the most informative portion of the gRNA sequences based methods is the region close to the PAM region, consistently across different datasets.

Then, the authors have investigated the high-activity preferential features within the PAM proximal regions reporting high intra specie similarities but no consistency across different species. From these results the authors conclude that the observed differences are driven by larger genomic sequence or context, that are specie dependent.

Briefly, this is a nice piece of work describing a simple yet ingenious analytical framework demonstrating the importance of the genomic context in determining gRNA efficiency and possibly explaining why all the algorithm designed so far to predict sgNRA efficiency from their sequence do not validate well on data coming from species that are different from those they have been trained on.

The manuscript is well written, the presented analyses are well conceived and results support authors' claim.

We thank the reviewer for these comments

I have few minor points, which I would like to see addressed or discussed in the response letter, before further considering this manuscript for publication.

1 - although the computational framework determining the most informative portions of the gRNA sequences is well designed and robustly cross-validated, the authors just used linear regression as predictive method, not even testing a single method of those presented in the publications from which the datasets are derived. Wouldn't be worthy to reperform the analysis including more methods?

We appreciate the reviewer's suggestion, but feel that while evaluating the impact of alternative algorithms is an interesting and worthy pursuit, this level of analysis is beyond the scope of this study. Our goal was to identify which part of the sequence most predicts activity and we believe a simple linear regression provides both the speed and general understandability that we needed to answer this specific question. More complicated algorithms such as SVMs, or other ML approaches, will highlight additional variables which may lead to significant insight within these data sets (specifically related to many of the additional confounding variables as discussed above in our response to Reviewer 1). For example, more complex neural networks may be able to capture complex relationships between sequences and activity, such as inhibitory secondary structure, that would not be captured in our analysis but that was not the intent of our analysis. While this is an intriguing set of studies we feel that this represents an orthogonal axis of information to the current analysis and feel the use of linear regression, which is simple and well understood by a more general audience is sufficient for the conclusions in the current study.

2 - several other (much larger datasets) of the same kind of those analysed on this manuscript are now on the public domain (for example those introduced in PMID: 29083409 and PMID: 30971826). Including these datasets in the analyses would be a nice plus

We appreciate this feedback. We have added these data being careful to remove redundancy. We have added a dataset from the first study (Meyers *et al* 2017, PMID: 29083409). In this study the same gRNA library was used across 342 different cell lines. Therefore to avoid redundancy, we selected gRNA activity data from one cell line and only gRNA that target

essential genes. There was a large amount of overlap between the gRNA targeting essential genes in the second study with the gRNA from the first study and so we didn't include those. However, we did include datasets from three other publications (PMID: 34050182, PMID: 27260156, PMID: 28162770). One of these is a large dataset targeting lentiviral integrated target sites and the other two datasets are similar to Meyers *et al.* In addition to these new Cas9 datasets, we also incorporated additional dCas9 and base editing datasets (discussed in Figure 6 and 7).

3 - the authors should briefly explain how gRNA activity is computed and how they account for gene targeted function? is the analysis restricted to 'essential genes' for example?

We apologize that this was not clear and have significantly updated our Methods Section (Page 24, Line 534) to discuss how gRNA activity was defined. Additionally, in Figure 2a as well as the text (Page 3, Line 87) we describe the method of gRNA activity measurement used by the studies authors. These include direct measurements that are sequencing based, essential phenotypic readouts (targeting essential genes), non-essential phenotype readouts, survival assays (mostly used in bacteria and yeast), and surveyor assay.

Reviewers' Comments:

Reviewer #1:

Remarks to the Author:

The authors have solidly addressed all my comments and concerns in their revised manuscript. For such, the study provides a very unique angle for better understanding how the PAM and PAM proximal sequences affect CRISPR-Cas9 gene editing activity. This study analyzed and benchmarked nearly all the published CRISPR gRNA activity datasets. It is a really plausible study. All figures and supplementary tables are nicely and clearly presented to support their findings and conclusions. And it is now suitable for publication and shared their analytic approach and findings within CRISPR society. A small correction to line 56-58 should be taken into consideration. The T-stretch signal is promoter specific, which should be corrected. E.g. U6 driven expression is affected with 4Ts or more. H1 and 7SK are from 5Ts. Refer to this pilot study: *Mol Ther Nucleic Acids*. 2018 Mar 2; 10: 36-44.

Reviewer #2:

Remarks to the Author:

The authors have satisfactorily addressed all the points I raised in the previous round of review.

Response to Reviewer & Editorial Comments:

Reviewer #1

The authors have solidly addressed all my comments and concerns in their revised manuscript. For such, the study provides a very unique angle for better understanding how the PAM and PAM proximal sequences affect CRISPR-Cas9 gene editing activity. This study analyzed and benchmarked nearly all the published CRISPR gRNA activity datasets. It is a really plausible study. All figures and supplementary tables are nicely and clearly presented to support their findings and conclusions. And it is now suitable for publication and shared their analytic approach and findings within CRISPR society. A small correction to line 56-58 should be taken into consideration. The T-stretch signal is promoter specific, which should be corrected. E.g. U6 driven expression is affected with 4Ts or more. H1 and 7SK are from 5Ts. Refer to this pilot study: Mol Ther Nucleic Acids. 2018 Mar 2; 10: 36–44.

We thank the reviewer and have made the following correction.

- a. Original sentence: One example is four contiguous thymines in a row, which is a transcriptional pause signal in some cell lines and organisms
 - b. New sentence: One example is four contiguous thymines in a row, which is a transcriptional pause signal when expressing gRNA from U6 promoters
-